# Active versus sham transcranial direct current stimulation (tDCS) as an adjunct to varenicline treatment for smoking cessation: Study protocol for a double-blind single dummy randomized controlled trial

**Laurie Zawertailo**[1,2,3]*, **Helena Zhang**[1,2], **Noreen Rahmani**[1,2], **Tarek K. Rajji**[3,4], **Peter Selby**[1,3,4,5,6]

**1** Nicotine Dependence Service, Addictions Program, Centre for Addiction and Mental Health, Toronto, Canada, **2** Department of Pharmacology and Toxicology, Faculty of Medicine, University of Toronto, Toronto, Canada, **3** Campbell Family Mental Health Research Institute, Centre for Addiction and Mental Health, Toronto, Canada, **4** Department of Psychiatry, University of Toronto, Toronto, Canada, **5** Department of Family and Community Medicine, Faculty of Medicine, University of Toronto, Toronto, Canada, **6** Dalla Lana School of Public Health, University of Toronto, Toronto, Canada

* laurie.zawertailo@camh.ca

## Abstract

### Background

Smoking is a chronic and relapsing disease, with up to 60% of quitters relapsing within the first year. Transcranial Direct Current Stimulation (tDCS), targets cortical circuits and acutely reduces craving and withdrawal symptoms among cigarette smokers. However, the efficacy of tDCS as an adjunct to standard smoking cessation treatments has not been studied. This study aims to investigate the effectiveness of tDCS in combination with varenicline for smoking cessation. We hypothesize that active tDCS combined with varenicline will improve cessation outcomes compared to sham tDCS combined with varenicline.

### Methods

This is a double-blind, sham-controlled randomized clinical trial where fifty healthy smokers will be recruited in Toronto, Canada. Participants will be randomized 1:1 to either active tDCS (20 minutes at 2 mA) or sham tDCS (30 seconds at 2 mA, 19 minutes at 0 mA) for 10 daily sessions (2 weeks) plus 5 follow up sessions, occurring every two weeks for 10 weeks. All participants will be given standard varenicline treatment concurrently for the 12-week treatment period. The primary outcome is 30 day continuous abstinence at end of treatment, confirmed with urinary cotinine. Measurements made at each study visit include expired carbon monoxide, self-reported craving and withdrawal. Three magnetic resonance imaging (MRI) scans will be conducted: two at baseline and one at end of treatment, to assess any functional or structural changes following treatment.

**Data Availability Statement:** The article does not report data, and the data availability policy is not applicable.

**Funding:** This research is funded by Global Research Awards for Nicotine Dependence (GRAND), a peer-reviewed research grant competition funded by Pfizer Inc (Zawertailo (GRAND2012) WS2391913).

**Competing interests:** All authors have completed the ICMJE uniform disclosure form at www.icmje.org/coi_disclosure.pdf and declare: no support from any organisation for the submitted work. PS reports receiving funding and/or honoraria from Pfizer Inc./Canada, Shoppers Drug Mart, Bhasin Consulting Fund Inc., Patient-Centered Outcomes Research Institute, ABBVie, and Bristol-Myers Squibb; LZ receives support from Pfizer Global Research Awards For Nicotine Dependence (GRAND) Award Program; there are no other relationships or activities that could appear to have influenced the submitted work. This does not alter our adherence to PLOS ONE policies on sharing data and materials. HZ and NR have no conflicts of interests to declare.

## Discussion

For every two smokers who quit, one life is saved from a tobacco-related mortality. Therefore, it is important to develop new and more effective treatment approaches that can improve and maintain long-term abstinence, in order to decrease the prevalence of tobacco-related deaths and disease. Furthermore, the addition of longitudinal neuroimaging can shed light on neural circuitry changes that might occur as a result of brain stimulation, furthering our understanding of tDCS in addiction treatment.

## Trial registration

This trial has been registered with Clinicaltrials.gov: NCT03841292 since February 15th 2019 (https://clinicaltrials.gov/ct2/show/NCT03841292)–retrospectively registered.

## 1. Background

Tobacco addiction is a chronic and relapsing condition, and one in five adults in North America continue to smoke despite knowing long-term harms [1]. While current first-line medications, such as nicotine replacement therapy (NRT), bupropion and varenicline, are moderately effective for cessation, relapse to smoking especially within the first six months of quitting is still the main barrier to full recovery [2]. Varenicline, an α4β2 nicotinic acetylcholine receptor partial agonist, is currently the most effective pharmacotherapy for smoking cessation with up to 50% abstinence by end of 12 weeks of treatment in clinical trials [3]. However, similar to other pharmacotherapies, successful long-term abstinence of varenicline is low, with approximately 40% relapsing by one year follow-up [4, 5].

Thus, there is still a great need and demand for new treatment options to increase successful quit attempts and decrease relapse. One of the main contributors to high relapse rates is succumbing to cravings elicited by environmental cues [6]. This vulnerability to relapse has been thought to be associated with long-term neuroadaptations that result from chronic cigarette smoking [6]. Neuroadaptation occurs with repeated chronic drug use, and often results in tolerance and dependence to the drug of choice. For smokers, neuroadaptation is also associated with desensitization of nicotinic acetylcholine receptors in the brain, sustaining nicotine dependence [7]. Under nicotine withdrawal, deficits in neuroplasticity have been implicated as the mechanism underlying the inability to decouple environmental cues and craving [8, 9].

Given the implication of neuroplasticity in substance use disorders, many emerging techniques have begun targeting neuroadaptations as a potential approach to treatment. Transcranial direct current stimulation (tDCS), a non-invasive brain stimulation technique, is emerging as a potential treatment for a variety of conditions such as schizophrenia, multiple sclerosis, Parkinson's disease, depression, [10–12] and addiction [13]. tDCS involves brief (10–20 min) application of weak electric current to the scalp [14] that increases the excitability of neurons under the anode. The procedure has been shown to be safe, convenient, fast-acting and with a well-established and dependable placebo manipulation ('sham' tDCS) [15]. tDCS has also been shown to modulate synaptic plasticity in human laboratory models of addictive motivation [16, 17].

While the mechanism of action of tDCS is not fully understood, one of the known mechanisms of tDCS is its ability to modify neuronal membrane polarity. Specifically, anodal tDCS decreases the threshold for action potential generation, by increasing the membrane

permeability to extracellular cations such as Na$^+$, Ca$^{2+}$ and K$^+$ [18, 19]. This modulation occurs mostly through the α4β2 and α7 ligand-gated nicotinic acetylcholine receptors, which are involved in neuroplasticity and modulation of cortical excitability [18, 19]. In lab-based experimental studies, tDCS decreases craving for cigarettes when the anode is placed over the left dorsolateral prefrontal cortex (DLPFC) [20, 21]. The DLPFC is implicated in neuroadaptations, controlling cravings and rewards related to smoking [22]. It is hypothesized that this increase in cortical excitability, specifically in the DLPFC, is the mechanism by which craving and cigarette consumption is decreased [23]. Since, varenicline and tDCS both act on α4β2 and α7 nicotinic acetylcholine receptors, using them in combination as a smoking cessation treatment could improve cessation outcomes above varenicline monotherapy.

Additionally, previous literature has shown an effect of non-invasive brain stimulation techniques on motor cortex excitability, indicating neural activity of these areas are also involved in the control of behavior. For example, a previous study demonstrated that when repetitive transcranial magnetic stimulation (rTMS), another non-invasive technique, was given over the right motor cortex (1 session), increased corticospinal excitability was seen during observation of happy and fearful emotional faces, compared to neutral ones [24]. The involvement of the motor cortex is especially interesting, seeing as emerging evidence has highlighted the importance of the motor circuitry in chronic nicotine use [25], specifically that chronic nicotine use is characterized by hyper-excitability of corticospinal output (a white matter tract that extends into the motor cortex). This increased excitability is speculated to be a secondary adaption to long-term nicotine use [25]. Additionally, previous literature has shown that tongue muscle motor-evoked potential (MEP) are sensitive to the neural processes that are activated during nicotine craving, suggesting a possible link between the corticobulbar pathway and reward pathways in smokers [26]. Collectively, these brain stimulation studies identified the relationship abnormal motor cortical excitability and nicotine dependence.

The mechanism of action of tDCS is still not fully understood. However, preclinical models have identified the ability of tDCS to affect polarization of neuronal membranes and glutamatergic plasticity [27, 28]. These effects involve spontaneous neuronal activity and can affect regional plasticity effects on cerebral networks. TDCS was shown to have effects on neural spiking and membrane potentials over a range of currents administered to rodents. Stochastic and rhythm resonance are the most plausible neural mechanisms by which weak modulation of an electric field affects neural information coding [29]. These modulations would manifest from small changes in spike predictability and timing, and could exert an effect of cognition via influence on neural population coding [29].

TDCS also affects long-term potentiation (LTP), another the mechanism that is implicated in the effects of tDCS on modulating brain function [30, 31]. Animal studies of nicotine dependence have shown the longer effect of direct current stimulation affecting LTP and that this effect was dependent on N-methyl D-aspartate and brain-derived neurotrophic factor (BDNF). More recent studies have shown that tDCS can also modulate presynaptic mechanisms of neuron signal transmission [32]. TDCS also has been shown to exert beneficial effects on neural plasticity and motor function in rodent models of stroke injuries, suggesting both neural and functional modulation potential [33]. Additionally, cigarette smokers have been shown to have increased brain activation in regions involved in cough sensory processing and cough suppression (such as the dorsolateral prefrontal cortex and midbrain nucleus cuneiformis) [34]. Previous literature also suggests that DLPFC tDCS stimulation may result in downstream effects such as subcortical dopamine release in other regions such as caudate nucleus [35], contributing to greater involvement of critical brain structures in reward and cognitive control. This suggests that tDCS can modulate deeper brain structures, contributing to its efficacy on reducing craving and smoking intake behaviors [34, 36].

To date, only one previous RCT has been conducted that compared bupropion (another first-line medication for smoking cessation) to tDCS stimulation as a treatment for smoking cessation [37]. In this sham-controlled randomized trial, 170 participants were recruited and randomized to receiving: A) 300 mg of bupropion over 12 weeks, B) 20 sessions of tDCS over 4 weeks, C) sham tDCS for 4 weeks, D) 20 sessions of tDCS over 12 weeks or E) sham tDCS over 12 weeks. End of treatment quit rates of participants using 300mg of bupropion was comparable to participants that received 20 sessions of left anodal tDCS (at 2 mA) over 12 weeks (20% versus 25.7% respectively). This was the first RCT to report therapeutic effects of tDCS on smoking cessation outcomes compared to a first-line pharmacotherapy [37], but did not combine pharmacotherapy with tDCS. Most studies to date have only examined the effects of tDCS on craving, motivation to smoke and smoking behavior measured in a laboratory setting rather than as a therapeutic intervention in treatment-seeking smokers.

Considering these gaps in literature, the primary objective of the study is to provide preliminary evidence of the efficacy of tDCS as an adjunct to 12 weeks of varenicline treatment for smoking cessation in a double-blind sham-controlled RCT. The secondary objective is to examine the neurobiological adaptations that may occur as a result of this treatment combination using functional magnetic resonance imaging (fMRI). We hypothesize that compared to sham tDCS plus varenicline, active tDCS to the left DLPFC in combination with varenicline treatment will increase both short and long-term quit rates and decrease relapse rates at end of treatment.

The relationship between smoking cue exposure and craving is often explored through cue reactivity paradigms in clinical neuroimaging studies [38]. In response to smoking-related cues, smokers have been shown to exhibit increased activation in regions involved in reward, craving emotions and memory, and visuospatial attention [39], suggesting increased attention to stimuli of heightened attentional salience. One study found that hyper-activation in some of these regions and decreased connectivity with cognitive control networks predicted "slips" in abstinence [6]. Behavioral and non-nicotinic pharmacological smoking interventions can normalize these brain responses relative to non-smoking controls [40, 41]. Thus, we hypothesize that active tDCS plus varenicline will reduce functional brain activation (measured by blood oxygenation level) to visual smoking cues.

We will also be conducting exploratory analysis on resting state functional connectivity, structural brain and diffusion tensor imaging data acquired during the MRI scans. Chronic cigarette smoking has been associated with various structural brain differences in areas such as the insula, prefrontal cortex and thalamus [42]. Resting state functional connectivity (rsFC) is a measure of blood oxygenation fluctuations between different brain regions during absence of task performance [43]. RsFC has been shown to be altered by chronic nicotine use in various brain areas in the mesolimbic system [44]. Lastly, using diffusion tensor imaging, a technique that measures structural white matter connectivity between brain regions, differences have been shown in white-matter connectivity between the prefrontal cortex and nucleus accumbens, corpus callosum and habenula in smokers compared to non-smokers [45]. We plan to explore, by a combination of functional and structural data, longitudinal activation patterns in response to active tDCS versus sham.

## 2. Methods

### 2.1 Study design overview

This study will be a double-blind, sham-controlled randomized controlled trial. A CONSORT diagram of the study and a schematic of study flow can be seen in Figs 1 and 2 respectively. Eligible participants will be randomly assigned (1:1) to active tDCS or sham tDCS. Prior to any

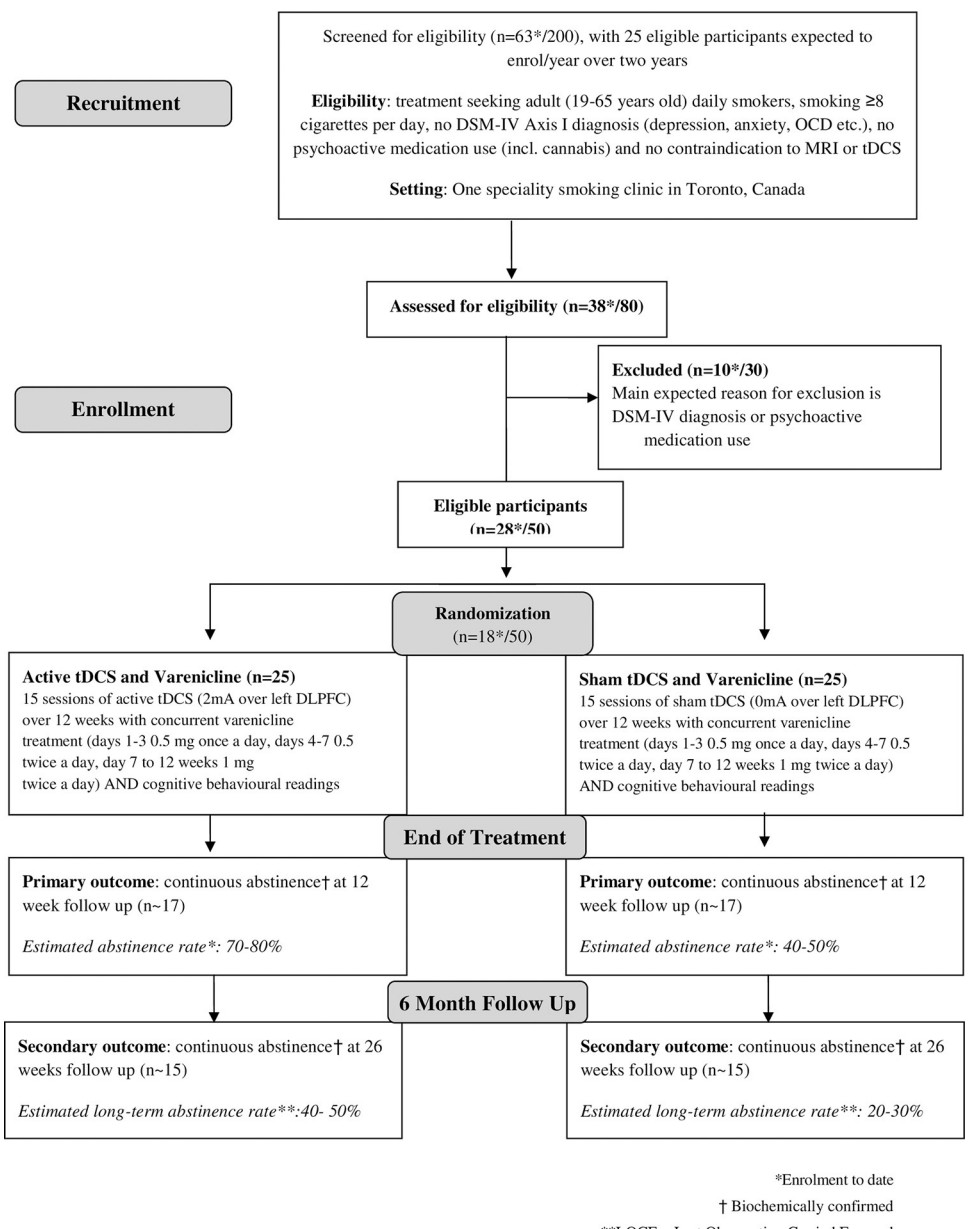

**Fig 1. Flow diagram of proposed study.**

treatment, participants will undergo two MRI scans, within 5 working days of each other. One scan will be conducted following overnight (at least 10-hours) smoking abstinence and the second baseline scan will be conducted within 1–2 hours of smoking a cigarette (satiated condition). During each scan, participants will be asked to complete a smoking cue-reactivity task, in which they will be shown a series of smoking-related, and content-matched neutral images. Once the baseline MRI scans are complete, participants will start the study treatment of 12 weeks of varenicline treatment, with concurrent tDCS stimulation sessions. On the same day that participants start taking varenicline, they will commence with 10 daily tDCS sessions over 2 weeks (Monday-Friday) followed by 'booster' session once every two weeks for the remainder of the 12-week course of varenicline. At the end of treatment, participants will undergo

| | Enrolment | Baseline scans Abs or Sat* | Baseline scans Abs or Sat* | Randomization | Daily** | Booster Sessions | Booster Sessions | Booster Sessions | Booster Sessions | Booster Sessions | EOT MRI scan | 6 month Follow up |
|---|---|---|---|---|---|---|---|---|---|---|---|---|
| **TIMEPOINT** | W1 | W1 | W2 | W3 | W3-4 | W6 | W8 | W10 | W12 | W14 (EOT) | W15 | W30 |
| **ENROLMENT:** | | | | | | | | | | | | |
| *Eligibility screen* | X | | | | | | | | | | | |
| *Informed consent* | X | | | | | | | | | | | |
| *Randomization* | | | | X | | | | | | | | |
| **MRI Scans** | | X | | | | | | | | | X | |
| **INTERVENTIONS:** | | | | | | | | | | | | |
| *Active 2mA tDCS + Varenicline* | | | | | ◆———————————————————◆ | | | | | | | |
| *Sham 0 mA tDCS +Varenicline* | | | | | ◆———————————————————◆ | | | | | | | |
| *Cognitive Behavioral readings* | | | | | ◆———————————————————◆ | | | | | | | |
| **ASSESSMENTS:** | | | | | | | | | | | | |
| *AUDIT, SSS, PANAS, EIS, State Trait, POMS* | X | | | | | | | | | X | | X |
| *QSU and FTND* | X | X | | | X | X | X | X | X | X | X | X |
| **OUTCOMES** | | | | | | | | | | | | |
| *30 Day Continuous Abstinence* | | | | | | | X | X | X | X | X | X |
| *Point Abstinence* | | | | | X | X | X | X | X | | X | X |
| *Cigarettes per day* | X | | | | X | X | X | X | X | X | | X |
| *BOLD activation* | | X | | | | | | | | | X | |
| **BIOCHEMICAL MEASURES** | | | | | | | | | | | | |
| *Urinary Cotinine* | | | | | | | | | | | X | X |
| *Pregnancy Test (if applicable)* | X | | | | | | | | | | | |
| *Urine drug screen* | | X | | | | | | | | | X | |
| *Expired carbon monoxide* | X | X | | | X | X | X | X | X | X | X | X |

**Fig 2. Schedule of enrolment interventions and assessments.** *Baseline MRI Scans will be counterbalanced such that participants will be undergoing an abstinence scan and a satiated scan within one week of each other. ** Daily tDCS stimulation over the first 2 weeks. List of abbreviations: MRI = functional magnetic resonance imaging; Eot = end of treatment; PHQ-9 = Patient Health Questionnaire–validated self-completed measure of depressive symptoms mapped on to DSM-IV criteria; QSU-brief = 10-item Questionnaire of Smoking Urges; AUDIT = Alcohol Use Disorders Identification Test; FTND = Fagerstrom Test for Nicotine Dependence; EIS = Eysenck Impulsivity Inventory; SSS = Sensation Seeking Scale; PANAS = Positive and Negative Affect Schedule; State Trait = State Trait Anxiety; POMS = Profile of Mood States.; BOLD = Blood oxygen level dependent signal, functional MRI measure of blood flow; Abs = MRI scan conducted following overnight nicotine abstinence (of at least 10 hours); Sat = MRI scan conducted following nicotine satiety (within 1–2 hours of smoking a cigarette).

another MRI scan under nicotine abstinence. Participants will return for a 6-month follow-up visit to assess their smoking status.

## 2.2 Study setting

The study will be conducted in the Nicotine Dependence Clinic within the Centre for Addiction and Mental Health, a tertiary care academic psychiatric hospital, fully affiliated with the University of Toronto, in downtown Toronto, Canada.

## 2.3 Participant recruitment

Participants will be recruited via online advertisements (Kijiji, Google Adwords Express and CAMH research registry), by word of mouth and poster bulletin boards within the community around the Greater Toronto Area.

## 2.4 Participants

Study participants will be fifty treatment-seeking daily dependent smokers.

**2.4.1 Ethics.** This protocol was approved by the Centre for Addiction and Mental Health Research Ethics Board (REB) under the protocol number 044–2016. Initial approval was obtained on November 14th 2016, the latest amendment was approved on March 6th 2018 with the version number 3.0. If any future amendments arise, all relevant parties will be notified.

## 2.5 Eligibility criteria

Eligible participants will be daily smokers between the ages of 19 to 65, smoking at least 8 cigarettes per day, treatment-seeking and willing to attend the required clinic appointments. Participants will be excluded if they: have any current DSM-IV Axis Diagnosis, except caffeine, gambling and phobias (assessed by M.I.N.I. SCID); current use of psychoactive drugs or medications; current use of other nicotine containing products (patches, electronic cigarettes, etc.); history of seizures or epilepsy; and/or life time history of concussions or head trauma. Additionally, according to standard protocols for tDCS clinical trials, participants will also be excluded if they: are pregnant or planning to become pregnant; have pacemakers or implanted electrical devices such as cochlear implants; have metal embedded in the skull; or have skin lesions, open wounds, bruising, or similar injuries at either of the stimulation sites.

Participants will also be excluded if they had contraindications to MRI, such as claustrophobia, a weight of more than 350 lbs, and previous experience working with grinding metal without protective eye equipment. Participants that have had a previous adverse reaction to varenicline will also be deemed ineligible for the study. When in treatment, participants will not be allowed to use any form of nicotine replacement product, e-cigarettes or drugs of abuse (including cannabis). Alcohol use was allowed in the study, as long as participants did not have an alcohol use disorder (assessed by the M.I.N.I. and AUDIT) and that their alcohol intake conforms with low-risk alcohol use guidelines (for men: <15 standardized drinks/week and for women: <10 standardized drinks/week) [46]. Participants reporting use of other nicotine products will be asked to stop, and if they cannot, they will be excluded from the trial. Participants will also be required to attend a physician visit, during which all risks and potential adverse events will be discussed with the Qualified Investigator (PS). The QI will also assess for potential risks for varenicline use, based on participants' concomitant medication and general health.

## 2.6 Prescreening

Interested participants will be screened by telephone. Potentially eligible participants will be invited for a baseline assessment visit for an in-person informed consent discussion and to confirm eligibility.

## 2.7 Baseline assessment

After signing an informed consent form (S2 Appendix), the baseline assessment will involve completion of a battery of questionnaires and the M.I.N.I. SCID for DSM-IV Axis I diagnosis. Baseline demographics of age, ethnicity, income, education, and smoking patterns will be collected for all participants prior to the start of treatment. Additionally, the following self-reports will be administered electronically, based on previous studies correlating these measures to substance use initiation and maintenance: the Fagerstrom Test for Nicotine Dependence [47], Questionnaire on Smoking Urges (QSU) [48], Positive and Negative Affect Schedule (PANAS) [49], Profile of Moods (POMS), Sensation Seeking Scale (SSS) [50], Alcohol Use Disorders Identification Test (AUDIT) [51] and the State Trait Anxiety Inventory. Please see the SPIRIT checklist (S1 Appendix) for all of the assessment and visit procedures.

## 2.8 Interventions

**2.8.1 Transcranial Direct Current Stimulation (tDCS).** The tDCS device used for simulation (SmartStim Model 1000; Nuraleve, Inc., Sudbury ON) is a battery-operated machine with a maximum 4 mA capacity (Nuraleve.com). TDCS will be applied via cutaneous anodal and cathodal electrodes– $35cm^2$ and $100cm^2$, respectively. Each electrode will consist of conductive carbon enclosed in a 0.9% NaCl saline-soaked sponge connected to lead wires attached to the tDCS device. An elasticized cap will hold the electrodes in place. The electrode configuration will be left DLPFC anodal tDCS following the standard montage successfully demonstrated in a number of tDCS addiction studies [16, 17, 20]. The anode and cathode will be applied to the F3 and F4 regions, respectively, as defined by the standard International EEG 10–20 system [52]. The tDCS machine will have software that is pre-programmed to randomize participants remotely, and provide stimulation without disclosing whether the session is active or sham. Thus, both participants and researchers will remain blinded throughout the study.

*2.8.1.1 Electric field simulation of tDCS montage.* To calculate the estimated electrical field received for each participant, the SimNibs pipeline will be used, established by the Danish Research Centre for Magnetic Resonance (DRCMR) and the Technical University of Denmark (DTU) [53]. To prepare the data for the stimulation, headreco was used to construct a tetrahedral head mesh from the T1 and T2 weighted structural MRI images [54].

Settings of the SimNibs for the anode electrode will be: 2.00 mA; F3 position (in subject space, calculated from head reconstruction); 3.5 by 5.2 cm electrode size (rectangle) with sponge and saline selected as the conductive (sponge = 5cm by 6 cm). For the cathode electrode, the settings will be: -2.00 mA, F4 position (in subject space), 12 by 7.6 cm electrode (rectangle) with sponge and saline selected as the conductive (sponge = 12.5 cm by 9.5 cm). An example of an estimated electric field calculation for a test participant is shown in **Fig 3**.

After the simulation is complete, the maximum electrical field (NormE) will be calculated for each participant. To calculate the electric field for the left DLPFC and right DLPFC respectively, MNI coordinates will be used to extract values from the mesh tetrahedron centers that was computed from the simulation. This will be completed using a custom MATLAB script, which is available on the SimNibs database. Lastly, the total electric field received during treatment for each participant will be calculated by multiplying the electric field of the left DLPFC

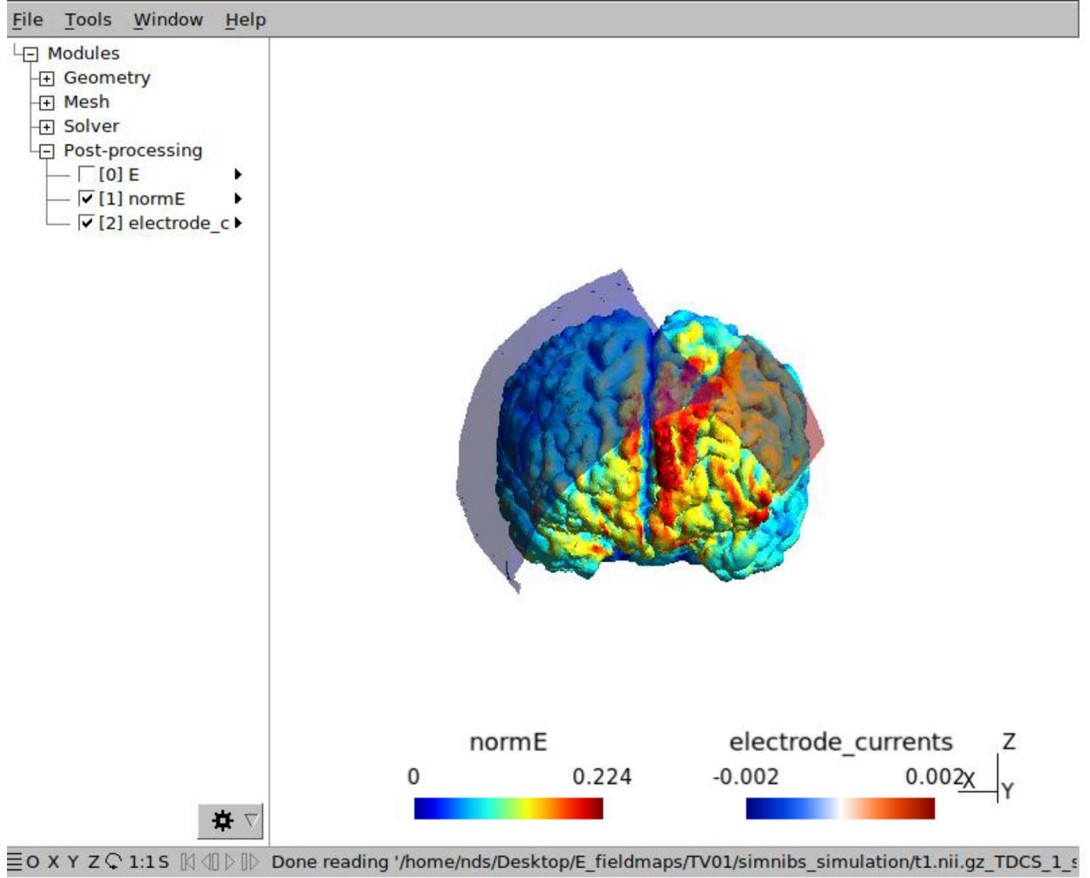

**Fig 3. Estimated electrical field for tDCS stimulation (using SimNIBs).**

by total number of tDCS sessions attended (max = 15). Electric fields will be compared between quitters versus non-quitters (amongst treatment completers), using two sample student's t-tests.

*2.8.1.2 Daily tDCS Sessions (week 1 to 2).* Each stimulation session will be 20 minutes long, conforming to parameters of a recent 5-session study on cigarette craving [16] and numerous other studies with clinical (depressed, substance dependence) and healthy participants [55]. Active treatment will consist of a current ramp up over 30 seconds to minimize discomfort, followed by 2 mA stimulation over 19 minutes and subsequent ramp down over the final 30 seconds. Sham treatment will include a 30-second ramp up and ramp down at the beginning of the session, 19 minutes of no stimulation and 30 second ramp up and down. The addition of the ramp up and ramp down in the sham treatment is to conceal the type of stimulation being received and maintain blinding to study condition, since tingling is commonly felt at the onset and termination. During each visit, participants' mood, smoking behavior, craving and withdrawal will be assessed and documented. Participants' quit status (if applicable at the time) will also be confirmed with expired carbon monoxide and urinary cotinine during these visits.

*2.8.1.3 Booster TDCS sessions (weeks 3 to 12).* After completing the ten daily stimulations, participants will return every two weeks for a booster session of tDCS (for a total of 5 boosters). The events of these stimulation sessions will be the same as daily sessions.

**2.8.2 Pharmacological treatment (weeks 1 through 12).** At the first tDCS session, all participants will start varenicline treatment for the duration of the 12 week treatment period.

Varenicline, as per standard protocol, will be prescribed with dose escalation: days 1 to 3, 0.5 mg once daily; days 4 to 7, 0.5 mg twice daily; days 8 to end of treatment, 1 mg twice daily. Medication will be used concurrently with tDCS sessions for a total of 12 weeks. Dose adjustments due to adverse events will be allowed (i.e. decrease to 0.5 mg twice daily). The target quit date for smoking cessation will be on the last day of the two weeks of daily tDCS, which corresponds to the recommended quit date after starting varenicline treatment. Participants will not be allowed to use any other nicotine products and psychoactive drugs during the treatment period. Participants will be asked to return their empty medication blister packs to assess for and encourage compliance. Any remaining pills will be recorded and then sent to CAMH Research Pharmacy to be destroyed.

**2.8.3 Cognitive behavioural reading material (weeks 1 through 12).** During each tDCS stimulation session, participants will be required to read self-help materials on cognitive behavioural strategies for smoking cessation, relapse prevention, and mood management. The material chosen originated from two cognitive behavioral therapy books: Quitting Smoking for Dummies [56] and Cognitive Behavioral Therapy for Dummies [57]. The readings for each day of stimulation will be standardized for all participants. The readings cover a wide array of areas such as how to deal with craving, recognizing craving, ways to maintain abstinence and healthy lifestyle habits. After each stimulation session, participants will be asked to fill in a brief questionnaire on their thoughts on the readings provided that session and what parts or changes they could incorporate into their daily routines. These readings will also serve the purpose of keeping participants attentive and engaged during each stimulation session.

## 2.9 Management of Adverse Events (AE) and Serious Adverse Events (SAE)

Participants will be asked at each study visit if they experience any adverse events (AE) from a list of the most prevalent side effects for tDCS and varenicline (S4 Appendix). If endorsed, participants will be asked to rate the side effect from mild to severe. The most commonly reported side effect of tDCS in a previously conducted trial are a mild tingling sensation (70.6%), moderate fatigue (35.3%), and a light itching sensation under the stimulation electrodes during stimulation (30.4%) [15]. After the session, the most common side effects were headache (11.8%), nausea (2.9%) and insomnia (0.98%). Overall, 17.7% of the volunteers experienced tDCS as mildly unpleasant. On the other hand, the most common side effects of varenicline are nausea, abnormal dreams, constipation, flatulence and vomiting in 30%, 13%, 8%, 6% and 5% of users, respectively [5, 58]. It is not known whether combining the two treatments will increase either the incidence or the severity of AEs, especially those that are common between the two treatments such as nausea and headache. If a participant experiences an adverse event that requires hospitalization or results in lasting harm or death (severe adverse event; SAE) and can reasonably be attributed to tDCS stimulation, the trial will be immediately suspended. AEs that can be expected based on the known profile of tDCS side effects will be managed by the Qualified Investigator (PS). Depending on the severity of the unexpected adverse event, the Qualified Investigator will decide if the trial should be suspended. All serious adverse events (SAE), adverse events (AEs) and unintentional adverse events (UAEs) will be reported to the CAMH Research Ethics Board. The emergency contact for reporting SAEs is the Qualified Investigator (PS).

## 2.10 Participant follow-up and retention

Participants will return to the clinic 3 months post end of treatment for a follow up visit, during which data will be collected regarding their craving, cigarette consumption and time to relapse (if applicable). Participants will also complete the same baseline behavioral assessments

(PANAS, QSU, etc.) during the follow-up. Participant retention will be encouraged through phone or email reminders of their upcoming appointments. Participants will also be compensated for their time during study participation as follows: 1) $ 70–80 CAD for each MRI scan, 2) $20 CAD + travel reimbursement ($6.50 for public transit) for each tDCS session and 3) $50 CAD for completion of the 6 month follow up visit. No additional data will be collected from participants who choose to drop out of the trial. Participants who lose contact, or miss more than 3 consecutive booster appointments will be considered drop-outs. To manage probable drop-outs, participants will be regularly contacted via email and phone of their upcoming appointments. Study staff will also be readily available to accommodate participant availabilities and/or changes to appointment schedules. Study staff will also be engaging in regular appointments to create a supportive and safe environment for participants. Lastly, varenicline will be dispensed at two time points (baseline and at 4 weeks follow up) to encourage treatment retention within the first month.

**2.10.1 Recruitment to date.** To date, 38 participants have been assessed for eligibility, of which 28 were deemed eligible (Fig 1). Of the 28 eligible participants, 26 participants successfully completed both baseline MRI scans and 18 participants were successfully randomized to either active or sham tDCS (and received at least one session of tDCS). Since the researchers are still blinded to the study conditions, we are unable to report how many participants are in each arm.

## 2.11 Neuroimaging

MRI scans will occur on 3 separate occasions: 1) at baseline following 10 hours of smoking abstinence (2) at baseline within 1–2 hours of smoking a cigarette and 3) at the end of the 12 week treatment. Baseline scans will occur within 1 week of each other and the order of satiated and abstinent scans will be counterbalanced to control for possible order effects. During each scan, we will measure functional activation while participants perform a smoking cue reactivity task (see cognitive tasks section), as well as structural data. During this task, participants will view smoking and neutral visual cues in a block design (each block = 20 seconds). A craving question will also be asked at the end of each cue block for 5 seconds (i.e. "how much do you want to smoke a cigarette right now"). The purpose of this of this question was two-fold: 1) to capture immediate nicotine craving that might be induced after each cue block presentation, and 2) to capture any delays in potential BOLD neural activity following cue presentation. Previous literature has shown peak BOLD response occurs 4 seconds after onset of a stimulus [59] and typically ends within 20 seconds [60, 61]. Thus, this question ensures that any potential delayed BOLD responses will be adequately captured after each block.

Lastly, each MRI scan will be preceded by an assessment of self-reported craving Questionnaire of Smoking Urges (QSU) [48]. Breathalyzer tests and urine will be collected prior to scans to confirm abstinence from alcohol and other drugs of abuse.

**2.11.1 Magnetic Resonance Imaging (MRI) acquisition parameters.** MRI will take place at the CAMH Research Imaging Centre with a GE Discovery MR750 3T Scanner (General Electric Medical Systems, Milwaukee, WI). Scans will be conducted using echo-planar imaging sequences (TE = 2650, 49 slices, whole brain, 3 mm isotropic voxels) and with a 32-channel head-coil for the smoking cue reactivity task (functional scan). Diffusion tensor imaging (DTI) will also be conducted with 30 directions. For structural scan acquisitions, T1 sequencing will be conducted with 0.9mm isotropic voxels. Resting state functional connectivity will be conducted using EPI sequences (TE = 2250, 42 slices, whole brain, 3.5 mm isotropic voxels).

**2.11.2 Smoking cue reactivity paradigm.** The design for the cue reactivity task can be seen in Fig 4, where participants will be presented with a series of smoking-related and neutral

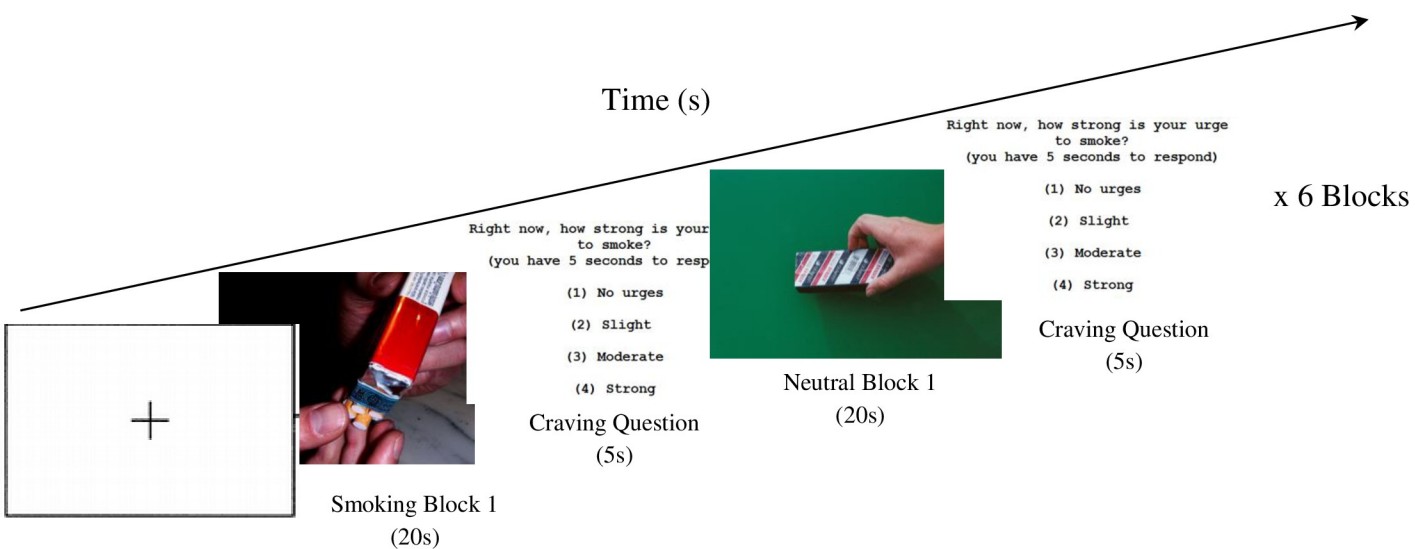

**Fig 4. Design for smoking cue reactivity task paradigm (MRI task).** During each MRI scan, participants will be presented images for approximately 7 minutes whilst being asked regarding their cigarette craving in between the blocks. The smoking block will consist of pictures related to smoking whilst the neutral block will have neutral images. Each block will last for 35 seconds for a total of 6 blocks. Blood oxygen level signal will be measured during this task. *Photos were taken from the Geneva Smoking Pictures (Khazaal et al. 2012).

photographic cues, adopted from the Geneva Smoking Pictures Series [62]. Smoking cues will include photographs of people smoking and smoking-related objects (i.e. packs of cigarettes, ashtrays, etc.). Neutral cues will consist of neutral themed pictures selected not to elicit strong emotional responses, such as neutral faces, flowers, or furniture. Neutral cues will be matched as closely as possible to smoking cues by color, depth, and positions. Visual cues will be presented in a block design, with 5 pictures per block, lasting a total of 20s. There will be 6 blocks each of neutral and smoking cues, presented in a random order, and all pictures presented within each block will be randomized. Each picture block will be interspersed with a rest fixation cross block lasting 10s. Between each block, participants will be asked how much they crave cigarettes on a scale of 1–4, with 4 being the highest craving.

### 2.12 Outcome measures

**2.12.1 Primary outcome.** The primary outcome measure of interest will be self-reported continuous abstinence from smoking during weeks 8–12 and confirmed using expired CO and urinary cotinine using a semi-quantitative dipstick (Rapid Response urinary cotinine stick) (S3 Appendix).

**2.12.2 Secondary outcomes.** Secondary outcome measures of interest will be the following:

1. Time to relapse, measured in weeks (if applicable) during the treatment period.

2. Self-reported continuous abstinence at week 26 confirmed with expired CO and urinary cotinine using a semi-quantitative dipstick.

3. Change in Blood Oxygen Level Dependent (BOLD) activation in response to smoking cues.

4. Explanatory outcome: comparing the structural brain (T1 and diffusion tensor imaging) and resting state functional connectivity between active and sham tDCS at end of treatment.

## 2.13 Sample size and power calculation

If we assume a 40% abstinence rate at end of treatment in the varenicline plus sham tDCS group, the abstinence rate in the active tDCS group would need to be double that or 80% in order to see a statistically significant difference with n = 25 per arm, p <0.05, and 80% power. Since this is a proof-of-concept study, for the treatment part of the study, the main purpose of the study will be to establish an effect size for tDCS as an adjunct to standard varenicline treatment. This is independent of the number of participants but allows us to estimate the N that would be required to achieve statistical significance depending on the effect size we observe. The effect size measure we will use will be the partial $n^2$ generated by SPSS. For the MRI portion of the study, 15 to 25 participants per group is typical in the neuroimaging field and is sufficient to detect significant within- and between-group differences in brain activity [63].

## 2.14 Randomization

Eligible participants will be randomized (1:1) to receiving active or sham tDCS treatment in blocks of 10. At baseline, the order of MRI scans will be counterbalanced, in which participants will be randomized in blocks of 10 to receiving the nicotine satiated scan first or the nicotine abstinent scan first.

## 2.15 Blinding

Both researchers and participants will be blinded to tDCS intervention. The tDCS device can program the type and duration of stimulation remotely so that the participant and all study staff will remain blinded to which group the participant is randomized to. The allocation sequence will be assigned and then administered by this software. The blind will be broken after the study has been completed. Participants will also be asked at end of treatment which treatment arm they think they were randomized to. To evaluate the effectiveness of the blind, we will investigate if participants were able to guess, better than chance, which stimulation they received.

## 2.16 Data collection

Baseline and follow up data will be collected using an electronic data collection platform (REDCap system) and urine samples will be collected for drug screening and confirmation of smoking abstinence (cotinine confirmed; S3 Appendix). Once enrolled in the study, participants will be assigned a unique identifier associated with their record with no other personal identifiers in order to maintain privacy. No personal health information will be stored on the electronic data platform.

## 2.17 Data management

Data will be entered by each participant as they complete a survey on an electronic data collection system. Data collected during the study will be stored on a secure network, which only research personnel can access. After participants enter their data, research assistants will check over the survey for missing items or duplicate entries. To improve data quality, the electronic surveys will also be designed to flag inappropriate values and ensure all values entered are in the correct metrics (minutes, days, years, etc.). For example, if participants indicate they smoke 300 cigarettes per day, the entry will be marked red and participants will not be able to submit their survey.

## 2.18 Data analysis

**2.18.1 Clinical data.** Intent-to-treat analysis will be conducted for the primary outcome, in which all participants who were successfully randomized and received at least one session of tDCS will be included. Participants that are lost to follow up during treatment will be considered non-quitter. The primary outcome of continuous abstinence (%) at end of treatment (12 weeks) will be compared between the active group versus the sham group using chi-square tests of independence. Binary logistic regression analysis will also be conducted with quit at end of treatment as the outcome variable and treatment condition (active or sham), as well as baseline variables as predictors. For example, cigarette consumption and dependence (FTND) will also be included as predictors in all binary regression models, to account for potential differences in addiction severity between the groups. For secondary outcomes, time to relapse (weeks or days if applicable) will be compared between the active versus the sham group using Kaplan-Meyer survival analysis for repeated measures. Similar to the primary outcomes, proportion of participants abstinent at 26 weeks follow up will be compared between active and sham tDCS using chi-square tests of independence and binary logistic regression with the same independent predictors.

**2.18.2 Neuroimaging data.** Brain imaging measures will be compared between treatment groups and across time points, using analysis of variance. MRI measures will also be related to subjective responses (i.e., baseline scores and changes in craving, mood, and withdrawal) using linear regression. fMRI data will be analyzed by fitting a general linear model (GLM) to the data time-series at every voxel across the brain (and/or within ROIs) and assessing effects using F/t-tests and percent BOLD signal change calculations using cluster-wise inference [64]. Paired t-tests will then be conducted to test for BOLD signal changes between active versus the sham group at end of treatment. Partial least squares (PLS) analysis will also be conducted to compare distributed BOLD signal differences across space and time in various brain regions in response to smoking cues versus neutral cues [65]. Latent variables derived from PLS will be compared between the active group versus the sham group at end of treatment. Resting state functional connectivity data will be analyzed using independence component analysis (ICA) for within-subject comparisons. Group differences between the active versus the sham group will be analyzed using tensorial independent component analysis, another form of a generalized linear model. For diffusion tensor imaging data, tract based spatial statistics will be used to compared fractional anisotropy differences between active versus sham and between quitters versus non-quitters respectively at end of treatment [66]. Lastly, structural brain data will be analyzed using FAST [67], enabling the comparisons of quantitative volumes of white matter, grey matter and cerebrospinal fluid respectively. Two sample student t-tests will be used to compare volumes of the above brain matters between active versus sham group at end of treatment.

## 3. Discussion

This study will be the first to investigate the effectiveness of adjunct tDCS treatment to varenicline for smoking cessation in treatment seeking smokers. Two previous studies have investigated the use of single-dose varenicline in conjunction with one session of tDCS in smokers and healthy controls to evaluate changes in cortical excitability in the human motor cortex [18, 19]. A 2018 review has also suggested that concurrent pharmacotherapy may impact the effect of tDCS on tissue excitability, especially when these medications influence neurotransmitter systems such as dopamine and serotonin [68].

However, this will be the first randomized clinical trial using the combination of tDCS with varenicline as a smoking cessation intervention for treatment seeking smokers. The proposed combination treatment has the potential to significantly increase efficacy of varenicline for

smoking cessation, decreasing disease burden and improving health outcomes for millions of smokers. TDCS is an easy, noninvasive and portable procedure and therefore, would be a convenient adjunct therapy to include in future smoking cessation interventions. The study is also a multi-modal imaging RCT, combining neuroimaging and clinical outcomes to give a more comprehensive picture on the mechanism of action of tDCS and mediated neural correlates that might be involved in varenicline treatment and smoking cessation.

Only a few studies have examined the effect of varenicline treatment on smoking cue reactivity. One study investigated the neural correlates of participants after taking 13 days of varenicline [69], and another study scanned participants after only 3 weeks of varenicline treatment [41]. There have been three previously conducted neuroimaging studies that investigated the effect of the full 12 week treatment of varenicline, but all three studies investigated baseline functional activation of a brain region or resting state functional connectivity [70–72]. This will be the first study that conducts a post-treatment scan, in a multi-modal manner, including both functional and structural data. Additionally, incorporation of longitudinal neuroimaging techniques to an intervention is scarcely done in the smoking literature and would add invaluable knowledge on both functional and structural brain correlates that could aid in identifying potential neural biomarkers of nicotine addiction or predictors of treatment outcome.

In light of the recent COVID19 pandemic, for future online or remote studies involving tDCS, utilization of commercially available home kits could offer new research opportunities and more potential participants. This home-based technique can be completed by participants with limited supervision, as seen by its use in other studies [73–75]. Similar to nicotine replacement therapy, varenicline can also be delivered in a remote setting via pharmacy or courier for future considerations [76].

## 3.1 Stimulation of the prefrontal cortex

The role of executive function deficits on addiction has been well-established in previous literature [22], making it a promising target for treatment potential using non-invasive brain stimulation techniques. Stimulation of the prefrontal cortex (PFC) is potentially linked to a plethora of neural correlates. This is because the PFC plays important and diverse roles in the assessment of rewards (both drug and non-drug) and the formation of reward-associated memories. The PFC can also retain reward information processing, and thus can be rapidly integrated with and update other somatosensory information, guiding behavior [77]. The PFC may also be an important contributor to the manifestation of goals, assignment of values to goals and the ability to select proper actions based on their value [77]. To respond to motivational salience and reward expectation, the PFC acts as a supervisor to other functions and may be down regulated in addiction [78].

Overall, the entire structure and all the sub-structures within the PFC contribute to the initiation of drug-seeking behaviors [79]. Previous research has also shown that cigarette smoking selectively impairs the prefrontal lobe [78], making it a desirable area to target in smoking cessation therapeutics. For example, lesions in the ventromedial prefrontal cortex (vmPFC) has been associated with poorer executive function, specifically, in acquisition of new learning outcomes and not just in reversal of previously learned behaviors [80]. Overall, the disruption of the prefrontal cortex function has been associated with behavioral disruptions such as value attribution, reward and anticipation [81].

## 3.2 Limitations

Firstly, due to COVID-19 disruptions, the final sample size may be smaller than planned initially. Secondly, the current study will not have a placebo varenicline group, and thus, it is not

possible to tease out the effect of the contribution of varenicline vs tDCS in this design. That being said, there are also distinct advantages of using the current design. For example, a previous critical review of trials comparing NIBs and pharmacotherapy concluded that the main advantage of a NIBs and pharmacotherapy trial is in allowing assessment of comparisons and combination effects of NIBs and pharmacotherapy, favoring these designs in future trials [82]. Although this is out of the scope of the current study, future studies can address this by inclusion of placebo varenicline groups (in a 2 by 2 factorial design), or by inclusion of other pharmacotherapies as adjunct therapies (i.e. bupropion or NRT with tDCS).

### 3.3 Future directions

The utility of brain stimulation adjunct therapies can be explored in many ways. Firstly, tDCS should be combined with other pharmacotherapy adjunct groups (bupropion and NRT). This will elucidate if the potential efficacy of the adjunct therapy is unique to varenicline, or if other pharmacotherapies can also benefit from the addition of tDCS. Other tDCS parameters can be explored in future studies to find the "sweet-spot" in terms of number of daily sessions, frequency of study visits, length of each stimulation and etc. This data would be important to inform future policy makers of an effective but also pragmatic treatment design. Additionally, instead of cognitive behavioral readings, it may be more beneficial to include in-person or virtual cognitive behavioral therapy in future designs to further equip participants with strategies and support to aid in their quit.

## 4. Conclusion

The risk of developing smoking-related diseases such as lung cancer and cardiovascular diseases decreases exponentially with duration of abstinence. As such, even a small improvement in the efficacy of intervention strategies can have a substantial population level effect, thereby decreasing the incidence of smoking and smoking-related diseases. Transcranial direct current stimulation (tDCS) has the potential to target underlying aberrant addiction neural circuitry; while varenicline has been shown to mediate smoking cessation via similar effects on the reward pathway. Combining the two interventions could result in improved cessation rates than either monotherapy, providing smokers with a more effective treatment option. Thus, findings from this novel adjunct tDCS and varenicline therapy will be clinically important, potentially decreasing the burden of smoking-related diseases.

## Supporting information

**S1 Appendix. SPIRIT checklist.**
(PDF)

**S2 Appendix. Informed consent.** Copy of the informed consent form.
(PDF)

**S3 Appendix. Biological specimens.** Details of biological specimen collection and the tests used for cotinine and drugs abuse.
(PDF)

**S4 Appendix. Adverse events of tDCS and varenicline.** Participants will complete the adverse event list at each study appointment. The list consists of the most common side effects of varenicline and tDCS respectively. Participants can indicate the severity by mild, moderate or severe.
(PDF)

**S1 File.**
(DOCX)

## Author Contributions

**Conceptualization:** Laurie Zawertailo, Tarek K. Rajji, Peter Selby.

**Methodology:** Laurie Zawertailo, Helena Zhang.

**Writing – original draft:** Laurie Zawertailo, Helena Zhang, Tarek K. Rajji, Peter Selby.

**Writing – review & editing:** Laurie Zawertailo, Helena Zhang, Noreen Rahmani, Tarek K. Rajji, Peter Selby.

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
