## [Decision Letter · Decision Letter 0]

20 Jun 2022

PONE-D-21-30439Active versus sham transcranial direct current stimulation (tDCS) as an adjunct to varenicline treatment for smoking cessation: study protocol for a double-blind single dummy randomized controlled trialPLOS ONE

Dear Dr. Zawertailo,

Thank you for submitting your manuscript to PLOS ONE. After careful consideration, we feel that it has merit but does not fully meet PLOS ONE’s publication criteria as it currently stands. Therefore, we invite you to submit a revised version of the manuscript that addresses the points raised during the review process.

Your manuscript has been assessed by three expert reviewers, whose comments are appended below. The reviewers have highlighted several areas where additional information or discussion would aid reproducibility and readability of your work. Please ensure you respond to each point carefully in your response to reviewers document, and modify your manuscript accordingly. We note that reviewer 1 has requested that you cite additional literature as part of your revisions. Please do review the recommendations and include any additional studies that you think are appropriate and add value to your manuscript, however there is no need to aim for any numerical target for total number of papers cited.

We look forward to receiving your revised manuscript.

Kind regards,

Joseph Donlan

Editorial Office

PLOS ONE

Journal Requirements:

3. Thank you for stating the following financial disclosure: "This research is funded by Global Research Awards for Nicotine Dependence (GRAND), a peer-reviewed research grant competition funded by Pfizer Inc (Zawertailo (GRAND2012) WS2391913). Pfizer Canada, the Canadian operation of Pfizer Inc, is located in Kirkland, Quebec, Canada (Telephone: 514-695-0500)."

Please state what role the funders took in the study.  If the funders had no role, please state: "The funders had no role in study design, data collection and analysis, decision to publish, or preparation of the manuscript.

4. Thank you for stating the following in the Acknowledgments Section of your manuscript: "Pfizer Inc. contribution consists of varenicline supply free of charge and funding obtained through the GRAND award program. "

Please remove any funding-related text from the manuscript and let us know how you would like to update your Funding Statement. Currently, your Funding Statement reads as follows: "This research is funded by Global Research Awards for Nicotine Dependence (GRAND), a peer-reviewed research grant competition funded by Pfizer Inc (Zawertailo (GRAND2012) WS2391913). Pfizer Canada, the Canadian operation of Pfizer Inc, is located in Kirkland, Quebec, Canada (Telephone: 514-695-0500)."

5. Thank you for stating the following in the Competing Interests section: "All authors have completed the ICMJE uniform disclosure form at www.icmje.org/coi_disclosure.pdf and declare: no support from any organisation for the submitted work. PS reports receiving funding and/or honoraria from Pfizer Inc./Canada, Shoppers Drug Mart, Bhasin Consulting Fund Inc., Patient-Centered Outcomes Research Institute, ABBVie, and Bristol-Myers Squibb; LZ receives support from Pfizer Global Research Awards in Nicotine dependence (GRAND) Award Program; there are no other relationships or activities that could appear to have influenced the submitted work. HZ and NR have no conflicts of interests to declare." 

We note that you received funding from a commercial sources: Pfizer Inc./Canada, Shoppers Drug Mart, Bhasin Consulting Fund Inc., ABBVie, and Bristol-Myers Squibb.

7. Please ensure that you refer to Figure 3 in your text as, if accepted, production will need this reference to link the reader to the figure.

9. We note that the original protocol file you uploaded contains a confidentiality notice indicating that the protocol may not be shared publicly or be published. Please note, however, that the PLOS Editorial Policy requires that the original protocol be published alongside your manuscript in the event of acceptance. Please note that should your paper be accepted, all content including the protocol will be published under the Creative Commons Attribution (CC BY) 4.0 license, which means that it will be freely available online, and any third party is permitted to access, download, copy, distribute, and use these materials in any way, even commercially, with proper attribution.

Therefore, we ask that you please seek permission from the study sponsor or body imposing the restriction on sharing this document to publish this protocol under CC BY 4.0 if your work is accepted. We kindly ask that you upload a formal statement signed by an institutional representative clarifying whether you will be able to comply with this policy. Additionally, please upload a clean copy of the protocol with the confidentiality notice (and any copyrighted institutional logos or signatures) removed.

10. We note that Figure 3 includes an image of a participant in the study. 

Reviewers' comments:

Reviewer's Responses to Questions

**Comments to the Author**

1. Does the manuscript provide a valid rationale for the proposed study, with clearly identified and justified research questions?

Reviewer #1: Yes

Reviewer #2: Partly

Reviewer #3: Yes

2. Is the protocol technically sound and planned in a manner that will lead to a meaningful outcome and allow testing the stated hypotheses?

Reviewer #1: Yes

Reviewer #2: Partly

Reviewer #3: Yes

3. Is the methodology feasible and described in sufficient detail to allow the work to be replicable?

Reviewer #1: Yes

Reviewer #2: Yes

Reviewer #3: Yes

4. Have the authors described where all data underlying the findings will be made available when the study is complete?

Reviewer #1: Yes

Reviewer #2: Yes

Reviewer #3: Yes

5. Is the manuscript presented in an intelligible fashion and written in standard English?

Reviewer #1: Yes

Reviewer #2: Yes

Reviewer #3: Yes

6. Review Comments to the Author

You may also provide optional suggestions and comments to authors that they might find helpful in planning their study.

Reviewer #1: Zawertailo and colleagues aim to investigate in the present study, entitled ‘Active versus sham transcranial direct current stimulation (tDCS) as an adjunct to varenicline treatment for smoking cessation: study protocol for a double-blind single dummy randomized controlled trial’, the current status of knowledge of the effectiveness of tDCS in combination with varenicline for smoking cessation. For this purpose, a double-blind, sham-controlled randomized clinical trial with fifty healthy will be conducted; participants will receive either active tDCS (20 minutes at 2 mA) or sham tDCS (30 seconds at 2 mA, 19.5 minutes at 0 mA) for 10 daily sessions plus 5 follow up sessions. All participants will be given standard varenicline treatment concurrently with stimulations. The primary outcome is a self-reported continuous abstinence from smoking, in addition to an improvement in smoking cessation due to tDCS-varenicline combined treatment.

The main strength of this paper is that it addresses an interesting and timely question, investigating the efficacy of tDCS as an adjunct to varenicline treatment for smoking cessation. In general, I think the idea of this article is really interesting and the authors’ fascinating observations on this timely topic may be of interest to the readers of Plos One. However, some comments, as well as some crucial evidence that should be included to support the authors’ argumentation, need to be addressed to improve the quality of the article, its adequacy, and its readability prior to the publication in the present form. My overall judgment is to publish this article after the authors have carefully considered my suggestions below, in particular reshaping the parts of the Introduction and Discussion sections.

Please consider the following comments:

• Regarding the Abstract: according to the Journal’s guidelines, authors should have provided an abstract of about 300 words maximum. Indeed, the current one includes 356 words. Please correct it.

• In general, I recommend authors to use more evidence to back their claims, especially in the Introduction of the article, which I believe is currently lacking. Thus, I recommend the authors to attempt to deepen the subject of their manuscript, as the bibliography is too concise: nonetheless, in my opinion, less than 70 articles for a research paper are really insufficient. Indeed, currently authors cite only 54 papers, and they are too low. Therefore, I suggest the authors to focus their efforts on researching more relevant literature: I believe that adding more studies and reviews will help them to provide better and more accurate background to this study. In this review, I will try to help the authors by suggesting some relevant literature of my knowledge that suit their manuscript.

• Introduction: The authors take a narrow view of the tDCS direct effect on cortical circuits underlying craving, describing its impact on symptoms among cigarette smokers in detail. Nevertheless, I think that a deeper examination of tDCS potential to modulate cortical excitability and, specifically, to induce motor cortical plasticity in humans would provide a better characterization and useful background in this instance. It is well known that non-invasive brain stimulation can modulate the increased neural activity of cortical areas involved in the control of behavior: in this regard, I would suggest adding evidence from a recent study that could provide interesting insights on the use of non-invasive brain stimulation (in this case, single-pulse TMS) to investigate the time course of the motor system readiness in response to processing of relevant arousing stimuli (i.e., happy and fearful faces), addressing how differences in the experience of aversive feelings modulate corticospinal excitability, hence, the preparation of adaptive motor responses required for the execution of appropriate behaviors (https://doi.org/10.3390/brainsci11091203). Results from this study, which involved healthy participants, helped gaining a better understating of the abnormal cortical excitability that characterizes craving. I also suggest an interesting article on altered excitability in chronic smokers (https://doi.org/10.1503/jpn.130086) that will offer a more coherent and defined background.

• Introduction: Following the first point raised, for a more thorough and comprehensive overview on this topic, I believe that it may be useful to have more information from additional evidence that focused on autonomic nervous system responses evoked by arousing stimuli (i.e., emotional stimuli or addiction-related stimuli). Importantly, recent findings suggest that responses to approaching emotional stimuli (i.e., emotional faces) modulate autonomic arousal as a function of the distance between the observer and the stimuli, resulting in an appropriate organization of defensive responses (https://doi.org/10.1007/s00221-020-05829-4). Similarly, results from another recent study showed that interpretation of potentially threatening situations, such as others’ proximity, triggers a number of physiological responses that help regulating the distance between ourselves and others during social interaction (https://doi.org/10.1038/s41598-021-82223-2). Moreover, authors might also see studies that have focused on this topic (https://doi.org/10.1152/ajpregu.00226.2018;
https://doi.org/10.1111/psyp.13636).

• Cognitive behavioural reading material (weeks 1 through 12): I suggest the authors to reorganize/reshape this section, providing detailed information about cognitive tests, as well as relevant scientific references, utilized in this session.

• Participant follow-up and retention: Could the authors discuss how they intend to manage probable drop-outs? Please explain it.

• Page 8, Discussion: In this section, authors thoroughly addressed their expected outcomes and their argumentation; however, I would have liked to see some views on a way forward. Hence, I ask them to include some thoughtful as well as in-depth considerations, making an effort, trying to explain the theoretical as well as the translational application of their research. In this respect, I believe that it could be useful to have more information from additional research that have focused on the pathophysiology underlying addiction, specifically on frontal cortex abnormalities and executive dysfunctions associated with impairments in memory and learning. Evidence from a recent study conducted on patients with lesion in the ventromedial portion of prefrontal cortex revealed that this brain structure is involved in the acquisition of emotional conditioning (i.e., learning) and assessed how naturally occurring bilateral lesion centered on the vmPFC compromises the generation of conditioned psychophysiological response during the acquisition of conditioning (https://doi.org/10.1523/JNEUROSCI.0304-20.2020). Accordingly, in a recent theoretical review that focused on neurobiology of fear conditioning, the distinct contributions of anterior/posterior subregions of the vmPFC in the processing of safety-threat information was discussed: here authors provided evidence for the fundamental role of this how region in the evaluation and representation of stimulus-outcome’s value needed to produce sustained physiological responses (https://doi.org/10.1038/s41380-021-01326-4). These findings highlight prefrontal cortex’s key role in the acquisition of new learning, and how its disrupted function may contribute to irregular behavioral responses and therefore to the development of neuropsychiatric disorders characterized by altered value attribution, reward anticipation or fear extinction (e.g., depression, anxiety, PTSD, substance-related and addictive disorders).

• Even though it is not mandatory, I believe that the ‘Conclusions’ section would be useful to adequately indicate convey what the authors believe will be the take-home message of their study, and therefore provide possible keys to advancing research and understanding of the prevalence of depression in post-stroke patients.

• In according to the previous comment, I would ask the authors to better define a ‘Limitations and future directions’ section before the end of the manuscript, in which authors can describe in detail and report all the technical issues that may be brought to the surface.

• Figures: I suggest to add a figure that displays stimulation sites’ position on the scalp. Also, I would appreciate some current flow modeling that could provide a better understanding of estimated current flow, for example using any of the free software like ROAST, COMETS, SimNIBS.

• References: According to the Journal’s guidelines, do not use a numbered list to include citations in the 'References' section.

Overall, the manuscript contains 3 figures and 54 references. In my opinion, the number of references it is too low for an original research article, and this issue may prevent the possibility of publishing it in this form. However, I believe that the manuscript may carry important value providing preliminary evidence of the efficacy of tDCS as an adjunct to 12 weeks of varenicline treatment for smoking cessation.

I hope that, after these careful revisions, the manuscript can meet the Journal’s high standards for publication. I am available for a new round of revision of this review.

I declare no conflict of interest regarding this manuscript.

Best regards,

Reviewer

Reviewer #2: The current manuscript is a protocol to conduct a study where the impact of transcranial direct current stimulation (tDCS) is studied as an adjunct therapy to varenicline for smoking cessation. The authors put a nice plan together to conduct this study, almost all is in place except two control groups are missing: (1) varenicline group alone and (2) tDCS group alone. Although the effect of these two groups have been reported previously, it is better to include these groups to enable the authors to make a firm conclusion regarding the combination therapy against the single therapy. The authors may argue that the varenicline plus sham tDCS group is the control for the current study. However, if this population do not respond to varenicline alone or tDCS alone, this information can be obtained by including these two additional groups. Also, the authors can make a firm conclusion whether the effect is additive or synergistic. Please also ensure to state that scans are taken during exposure to neutral cues as well as smoking cue to ensure the scans can be correlated to the level of craving. One issue with this approach is that if neuronal connectivity or activation occurs in a delayed manner and during the time the other cues are shown, how the authors can interpret that and differentiate neuronal activation associated with the cues if the signal shows up after the exposure to the cue. Some parts of the manuscripts are written in the future tense and others in past tense. Please ensure that tense is used appropriate for a given sentence. Also, change the following minor typos.

Minor:

1. Please change "nicotine" to "nicotinic" on line 81.

2. Please use the same abbreviation for both lines 144 and 146. I suggest to use RsFC in both lines.

3. Please change to "differences have been" on line 148.

4. The word :respectively" may not be needed on line 149.

5. Please leave a comma between QSU and etc. on line 304.

6. Please change to "at the end of the 12-week treatment" on line 321.

7. Please delete the description about BOLD since this was defined above (line 362).

8. Please change "at end of the treatment" to " at the end of the treatment" throughout the manuscript.

9. Please leave a comma between years and etc. on line 406.

10. Please change "loss" to "lost" and "not quit" to "no-quitter" on line 415.

Reviewer #3: PONE-D-21-30439

Active versus sham transcranial direct current stimulation (tDCS) as an adjunct to varenicline treatment for smoking cessation: study protocol for a double-blind single dummy randomized controlled trial.

General Comments:

This manuscript describes a study protocol for a double-blind RCT examining the effectiveness of tDCS as an adjunctive treatment with varenicline pharmacotherapy (varenicline is a alpha4beta2 nicotinic ligand and is a form of nicotine replacement therapy).

The authors do a good job of establishing the rationale for this study. The effectiveness of pharmacotherapy alone treatment in smoking cessation treatment is limited and thus, examining potential adjunctive treatments (drug, physiological, or behavioral) that may increase long term smoking cessation success is an important and clinically relevant research endeavor.

The methodology is sound and uses best practices of RCT plus sham control groups. Baseline MRI scans will permit determination of changes due to treatment in from both nicotine-deprivation and -satiation conditions. The 12-wk treatment protocol should allow for determination of overall treatment effectiveness. The researchers propose to assess multiple behavioral measures (mood, craving, withdrawal) at each visit and importantly, will also use biological measures of CO ppm and urinary continine levels to verify smoking status. The authors have chosen appropriate doses and employed a standard dose-escalation procedure. As this is a proof-of-concept study to better establish experimental parameters for future work, these methodological details seem appropriate.

Significance: as there is sparse literature on the potential for tDCS to enhance currently approved pharmacotherapies for smoking cessation is great. This protocol has the potential to reveal new avenues for treatments to improve smoking cessation paradigms.

Comments and/or Points for the authors to consider:

1. Current knowledge suggests individualized treatment for smoking cessation is necessary – and may largely be a function of individual experience, history, dependence severity, etc. The authors describe inclusion criteria of at least 8 cigarettes per day, which strikes this reader as setting the bar relatively low and perhaps creating a more heterogeneous sample (e.g., 8 cigarettes/day for 1 year vs 30 cigarettes/day for 10 years would be two very different users in my experience). Thus, this variable sample may hinder the ability to find significant differences. However, if the authors matched their groups on this factor, or somehow controlled for severity statistically, that may mitigate some of these methodological concerns.

2. Exclusion criteria include other drugs of abuse and the authors specifically mention cannabis products but it is not clear if this definition also includes alcohol. This is important because varenicline has interactions with alcohol, thus patients should be at least cautioned about concurrent alcohol use. Varenicline also increases risk for adverse reactions in individuals with renal disease and it’s unclear if subjects will be screened for that.

7. PLOS authors have the option to publish the peer review history of their article (what does this mean?). If published, this will include your full peer review and any attached files.

Reviewer #1: No

Reviewer #2: **Yes: **Kabirullah Lutfy

Reviewer #3: No

---

## [Author Response · Author response to Decision Letter 0]

9 Aug 2022

Response: File naming has been updated (p. 37). 

Response: Updated Financial Disclosure. 

3. Thank you for stating the following financial disclosure: "This research is funded by Global Research Awards for Nicotine Dependence (GRAND), a peer-reviewed research grant competition funded by Pfizer Inc (Zawertailo (GRAND2012) WS2391913). Pfizer Canada, the Canadian operation of Pfizer Inc, is located in Kirkland, Quebec, Canada (Telephone: 514-695-0500)."

Please state what role the funders took in the study. If the funders had no role, please state: "The funders had no role in study design, data collection and analysis, decision to publish, or preparation of the manuscript.

Response: The statement has been added in the cover letter and on the electronic submission. 

4. Thank you for stating the following in the Acknowledgments Section of your manuscript: "Pfizer Inc. contribution consists of varenicline supply free of charge and funding obtained through the GRAND award program."

Please remove any funding-related text from the manuscript and let us know how you would like to update your Funding Statement. Currently, your Funding Statement reads as follows: "This research is funded by Global Research Awards for Nicotine Dependence (GRAND), a peer-reviewed research grant competition funded by Pfizer Inc (Zawertailo (GRAND2012) WS2391913). Pfizer Canada, the Canadian operation of Pfizer Inc, is located in Kirkland, Quebec, Canada (Telephone: 514-695-0500)."

Response: Removed. 

5. Thank you for stating the following in the Competing Interests section: "All authors have completed the ICMJE uniform disclosure form at www.icmje.org/coi_disclosure.pdf and declare: no support from any organisation for the submitted work. PS reports receiving funding and/or honoraria from Pfizer Inc./Canada, Shoppers Drug Mart, Bhasin Consulting Fund Inc., Patient-Centered Outcomes Research Institute, ABBVie, and Bristol-Myers Squibb; LZ receives support from Pfizer Global Research Awards in Nicotine dependence (GRAND) Award Program; there are no other relationships or activities that could appear to have influenced the submitted work. HZ and NR have no conflicts of interests to declare." 

We note that you received funding from a commercial sources: Pfizer Inc./Canada, Shoppers Drug Mart, Bhasin Consulting Fund Inc., ABBVie, and Bristol-Myers Squibb.

Response: This has been amended (p.30) and in the new cover letter submitted. 

Response: Updated to show in Methods (p. 11)

7. Please ensure that you refer to Figure 3 in your text as, if accepted, production will need this reference to link the reader to the figure.

Response: Addressed (p.19). Note: Figure 3 is now Figure 4, due to the addition of a new Figure.

Response: Captions for supporting information files have been added at the end (p.37). 

9. We note that the original protocol file you uploaded contains a confidentiality notice indicating that the protocol may not be shared publicly or be published. Please note, however, that the PLOS Editorial Policy requires that the original protocol be published alongside your manuscript in the event of acceptance. Please note that should your paper be accepted, all content including the protocol will be published under the Creative Commons Attribution (CC BY) 4.0 license, which means that it will be freely available online, and any third party is permitted to access, download, copy, distribute, and use these materials in any way, even commercially, with proper attribution.

Therefore, we ask that you please seek permission from the study sponsor or body imposing the restriction on sharing this document to publish this protocol under CC BY 4.0 if your work is accepted. We kindly ask that you upload a formal statement signed by an institutional representative clarifying whether you will be able to comply with this policy. Additionally, please upload a clean copy of the protocol with the confidentiality notice (and any copyrighted institutional logos or signatures) removed.

Response: Updated on p. 31: “This protocol will be granted public access”. Study protocol will be made available. There was no sponsor or body imposing this restriction on document sharing. 

10. We note that Figure 3 includes an image of a participant in the study. 

Response: Picture has been replaced with a photo with no identifying or potentially identifying information. We have added the source of the photos to the figure caption (p.36).

6. Review Comments to the Author

Reviewer #1: Zawertailo and colleagues aim to investigate in the present study, entitled ‘Active versus sham transcranial direct current stimulation (tDCS) as an adjunct to varenicline treatment for smoking cessation: study protocol for a double-blind single dummy randomized controlled trial’, the current status of knowledge of the effectiveness of tDCS in combination with varenicline for smoking cessation. For this purpose, a double-blind, sham-controlled randomized clinical trial with fifty healthy will be conducted; participants will receive either active tDCS (20 minutes at 2 mA) or sham tDCS (30 seconds at 2 mA, 19.5 minutes at 0 mA) for 10 daily sessions plus 5 follow up sessions. All participants will be given standard varenicline treatment concurrently with stimulations. The primary outcome is a self-reported continuous abstinence from smoking, in addition to an improvement in smoking cessation due to tDCS-varenicline combined treatment. The main strength of this paper is that it addresses an interesting and timely question, investigating the efficacy of tDCS as an adjunct to varenicline treatment for smoking cessation. In general, I think the idea of this article is really interesting and the authors’ fascinating observations on this timely topic may be of interest to the readers of Plos One. However, some comments, as well as some crucial evidence that should be included to support the authors’ argumentation, need to be addressed to improve the quality of the article, its adequacy, and its readability prior to the publication in the present form. My overall judgment is to publish this article after the authors have carefully considered my suggestions below, in particular reshaping the parts of the Introduction and Discussion sections.

Please consider the following comments:

• Regarding the Abstract: according to the Journal’s guidelines, authors should have provided an abstract of about 300 words maximum. Indeed, the current one includes 356 words. Please correct it.

Response: Thank you for your review and helpful suggestions. This has been addressed (Abstract is now 299 words, p.3)

• In general, I recommend authors to use more evidence to back their claims, especially in the Introduction of the article, which I believe is currently lacking. Thus, I recommend the authors to attempt to deepen the subject of their manuscript, as the bibliography is too concise: nonetheless, in my opinion, less than 70 articles for a research paper are really insufficient. Indeed, currently authors cite only 54 papers, and they are too low. Therefore, I suggest the authors to focus their efforts on researching more relevant literature: I believe that adding more studies and reviews will help them to provide better and more accurate background to this study. In this review, I will try to help the authors by suggesting some relevant literature of my knowledge that suit their manuscript.

Response: Addressed in manuscript. There are now 79 references in the manuscript. 

• Introduction: The authors take a narrow view of the tDCS direct effect on cortical circuits underlying craving, describing its impact on symptoms among cigarette smokers in detail. Nevertheless, I think that a deeper examination of tDCS potential to modulate cortical excitability and, specifically, to induce motor cortical plasticity in humans would provide a better characterization and useful background in this instance. It is well known that non-invasive brain stimulation can modulate the increased neural activity of cortical areas involved in the control of behavior: in this regard, I would suggest adding evidence from a recent study that could provide interesting insights on the use of non-invasive brain stimulation (in this case, single-pulse TMS) to investigate the time course of the motor system readiness in response to processing of relevant arousing stimuli (i.e., happy and fearful faces), addressing how differences in the experience of aversive feelings modulate corticospinal excitability, hence, the preparation of adaptive motor responses required for the execution of appropriate behaviors (https://doi.org/10.3390/brainsci11091203). Results from this study, which involved healthy participants, helped gaining a better understating of the abnormal cortical excitability that characterizes craving. I also suggest an interesting article on altered excitability in chronic smokers (https://doi.org/10.1503/jpn.130086) that will offer a more coherent and defined background.

Response: Thank you for the comment and article suggestions. This has been added in the introduction (p.7-8) 

• Introduction: Following the first point raised, for a more thorough and comprehensive overview on this topic, I believe that it may be useful to have more information from additional evidence that focused on autonomic nervous system responses evoked by arousing stimuli (i.e., emotional stimuli or addiction-related stimuli). Importantly, recent findings suggest that responses to approaching emotional stimuli (i.e., emotional faces) modulate autonomic arousal as a function of the distance between the observer and the stimuli, resulting in an appropriate organization of defensive responses (https://doi.org/10.1007/s00221-020-05829-4). Similarly, results from another recent study showed that interpretation of potentially threatening situations, such as others’ proximity, triggers a number of physiological responses that help regulating the distance between ourselves and others during social interaction (https://doi.org/10.1038/s41598-021-82223-2). Moreover, authors might also see studies that have focused on this topic (https://doi.org/10.1152/ajpregu.00226.2018;
https://doi.org/10.1111/psyp.13636).

Response: Thank you for the suggestion, however, while these are interesting papers, we feel they do not pertain to our protocol studying tDCS as a treatment for nicotine dependence. 

• Cognitive behavioural reading material (weeks 1 through 12): I suggest the authors to reorganize/reshape this section, providing detailed information about cognitive tests, as well as relevant scientific references, utilized in this session.

Response: Added in cognitive behavioral section (p.16-17). There were no cognitive tests performed during or after tDCS sessions. References for the reading materials have been added. 

• Participant follow-up and retention: Could the authors discuss how they intend to manage probable drop-outs? Please explain it.

Response: Added in methods (p.18): Since this is a proof-of-concept trial, participants that lose contact, or miss more than 3 consecutive booster sessions will be considered as drop-outs. To manage probable drop-outs, participants will be regularly contacted via email and phone of their upcoming appointments. Study staff will also be readily available to accommodate participant availabilities and/or changes to appointment schedules. Study staff will also be engaging in regular appointments to create a supportive and safe environment for participants. Lastly, varenicline will be dispensed at two time points (baseline and at 4 weeks follow up) to encourage treatment retention within the first month.

• Page 8, Discussion: In this section, authors thoroughly addressed their expected outcomes and their argumentation; however, I would have liked to see some views on a way forward. Hence, I ask them to include some thoughtful as well as in-depth considerations, making an effort, trying to explain the theoretical as well as the translational application of their research. In this respect, I believe that it could be useful to have more information from additional research that have focused on the pathophysiology underlying addiction, specifically on frontal cortex abnormalities and executive dysfunctions associated with impairments in memory and learning. Evidence from a recent study conducted on patients with lesion in the ventromedial portion of prefrontal cortex revealed that this brain structure is involved in the acquisition of emotional conditioning (i.e., learning) and assessed how naturally occurring bilateral lesion centered on the vmPFC compromises the generation of conditioned psychophysiological response during the acquisition of conditioning (https://doi.org/10.1523/JNEUROSCI.0304-20.2020). Accordingly, in a recent theoretical review that focused on neurobiology of fear conditioning, the distinct contributions of anterior/posterior subregions of the vmPFC in the processing of safety-threat information was discussed: here authors provided evidence for the fundamental role of this how region in the evaluation and representation of stimulus-outcome’s value needed to produce sustained physiological responses (https://doi.org/10.1038/s41380-021-01326-4). These findings highlight prefrontal cortex’s key role in the acquisition of new learning, and how its disrupted function may contribute to irregular behavioral responses and therefore to the development of neuropsychiatric disorders characterized by altered value attribution, reward anticipation or fear extinction (e.g., depression, anxiety, PTSD, substance-related and addictive disorders).

Response: Thank you for this suggestion. Additional information has been added in the discussion (p.26-28)

• Even though it is not mandatory, I believe that the ‘Conclusions’ section would be useful to adequately indicate convey what the authors believe will be the take-home message of their study, and therefore provide possible keys to advancing research and understanding of the prevalence of depression in post-stroke patients.

Response: Addressed, added Conclusion statement (p. 28-29)

• In according to the previous comment, I would ask the authors to better define a ‘Limitations and future directions’ section before the end of the manuscript, in which authors can describe in detail and report all the technical issues that may be brought to the surface.

Response: Addressed, added a Limitations and Future Direction sections (p.28).

• Figures: I suggest to add a figure that displays stimulation sites’ position on the scalp. Also, I would appreciate some current flow modeling that could provide a better understanding of estimated current flow, for example using any of the free software like ROAST, COMETS, SimNIBS.

Response: Electrical field calculations are now addressed in methods (p.14-15) and a new figure (Figure 3) has been added to show the estimated current flow using SimNIBs of a test participant. 

• References: According to the Journal’s guidelines, do not use a numbered list to include citations in the 'References' section.

Response: According to the PLOS One submission guidelines, references should be in Vancouver style and numbered in order of appearance. 

Overall, the manuscript contains 3 figures and 54 references. In my opinion, the number of references it is too low for an original research article, and this issue may prevent the possibility of publishing it in this form. However, I believe that the manuscript may carry important value providing preliminary evidence of the efficacy of tDCS as an adjunct to 12 weeks of varenicline treatment for smoking cessation. I hope that, after these careful revisions, the manuscript can meet the Journal’s high standards for publication. I am available for a new round of revision of this review. I declare no conflict of interest regarding this manuscript.

Best regards,

Reviewer

Response: The revised manuscript now contains 4 figures and 79 references. A more in-depth introduction has been added and the discussion has been expanded as well. Thank you!

Reviewer #2: The current manuscript is a protocol to conduct a study where the impact of transcranial direct current stimulation (tDCS) is studied as an adjunct therapy to varenicline for smoking cessation. The authors put a nice plan together to conduct this study, almost all is in place except two control groups are missing: (1) varenicline group alone and (2) tDCS group alone. Although the effect of these two groups have been reported previously, it is better to include these groups to enable the authors to make a firm conclusion regarding the combination therapy against the single therapy. The authors may argue that the varenicline plus sham tDCS group is the control for the current study. However, if this population do not respond to varenicline alone or tDCS alone, this information can be obtained by including these two additional groups. Also, the authors can make a firm conclusion whether the effect is additive or synergistic. Please also ensure to state that scans are taken during exposure to neutral cues as well as smoking cue to ensure the scans can be correlated to the level of craving. One issue with this approach is that if neuronal connectivity or activation occurs in a delayed manner and during the time the other cues are shown, how the authors can interpret that and differentiate neuronal activation associated with the cues if the signal shows up after the exposure to the cue. Some parts of the manuscripts are written in the future tense and others in past tense. Please ensure that tense is used appropriate for a given sentence. Also, change the following minor typos.

Response: Thank you for your review! Regarding the placebo varenicline group, this is limitation is now addressed in the discussion section (p.27-28). In regards to the MRI scans, a sentence has been added to the methods section for the state that the scans are taken and for the potential delay of BOLD signal response and how the authors plan on addressing this (p.20). The tense has also been changed in some parts to be future tense. 

Minor:

1. Please change "nicotine" to "nicotinic" on line 81.

2. Please use the same abbreviation for both lines 144 and 146. I suggest to use RsFC in both lines.

3. Please change to "differences have been" on line 148.

4. The word: respectively" may not be needed on line 149.

5. Please leave a comma between QSU and etc. on line 304.

6. Please change to "at the end of the 12-week treatment" on line 321.

7. Please delete the description about BOLD since this was defined above (line 362).

8. Please change "at end of the treatment" to " at the end of the treatment" throughout the manuscript.

9. Please leave a comma between years and etc. on line 406.

10. Please change "loss" to "lost" and "not quit" to "no-quitter" on line 415.

Response: All minor comments have been addressed in the manuscript. 

Reviewer #3: PONE-D-21-30439

Active versus sham transcranial direct current stimulation (tDCS) as an adjunct to varenicline treatment for smoking cessation: study protocol for a double-blind single dummy randomized controlled trial.

General Comments:

This manuscript describes a study protocol for a double-blind RCT examining the effectiveness of tDCS as an adjunctive treatment with varenicline pharmacotherapy (varenicline is a alpha4beta2 nicotinic ligand and is a form of nicotine replacement therapy).

The authors do a good job of establishing the rationale for this study. The effectiveness of pharmacotherapy alone treatment in smoking cessation treatment is limited and thus, examining potential adjunctive treatments (drug, physiological, or behavioral) that may increase long term smoking cessation success is an important and clinically relevant research endeavor.

The methodology is sound and uses best practices of RCT plus sham control groups. Baseline MRI scans will permit determination of changes due to treatment in from both nicotine-deprivation and -satiation conditions. The 12-wk treatment protocol should allow for determination of overall treatment effectiveness. The researchers propose to assess multiple behavioral measures (mood, craving, withdrawal) at each visit and importantly, will also use biological measures of CO ppm and urinary continine levels to verify smoking status. The authors have chosen appropriate doses and employed a standard dose-escalation procedure. As this is a proof-of-concept study to better establish experimental parameters for future work, these methodological details seem appropriate.

Significance: as there is sparse literature on the potential for tDCS to enhance currently approved pharmacotherapies for smoking cessation is great. This protocol has the potential to reveal new avenues for treatments to improve smoking cessation paradigms.

Comments and/or Points for the authors to consider:

1. Current knowledge suggests individualized treatment for smoking cessation is necessary – and may largely be a function of individual experience, history, dependence severity, etc. The authors describe inclusion criteria of at least 8 cigarettes per day, which strikes this reader as setting the bar relatively low and perhaps creating a more heterogeneous sample (e.g., 8 cigarettes/day for 1 year vs 30 cigarettes/day for 10 years would be two very different users in my experience). Thus, this variable sample may hinder the ability to find significant differences. However, if the authors matched their groups on this factor, or somehow controlled for severity statistically, that may mitigate some of these methodological concerns.

Response: Thank you for the review and comments. The rationale for including smokers with 8 cigarettes per day is based on the observation that 1) average cigarette consumption of a moderate smoker is between 10-19 cigarettes per day and 2) epidemiological data has consistently shown that smokers tend to underreport their cigarette consumption and therefore a more conservative cut-off was used to encourage recruitment of a sample population that is representative of the general population of smokers in Canada. That being said, the authors do plan on controlling for cigarette consumption and nicotine dependence in statistical analyses (added on p. 24). 

2. Exclusion criteria include other drugs of abuse and the authors specifically mention cannabis products but it is not clear if this definition also includes alcohol. This is important because varenicline has interactions with alcohol, thus patients should be at least cautioned about concurrent alcohol use. Varenicline also increases risk for adverse reactions in individuals with renal disease and it’s unclear if subjects will be screened for that.

Response: Alcohol use will be allowed in the study, assuming that participants do not have an alcohol use disorder (assessed by the M.I.N.I and AUDIT) and are drinking within Canada’s low-risk alcohol drinking guidelines (for women: no more than 10 standard drinks/week, for men:15 drinks/week). A section has been added in Methods (p. 12). Participants will also be required to attend a physician visit, during which the qualified investigator (P.S.) will assess for potential risks for participants and decide if varenicline would be safe for the participant. Participants will also be cautioned in this appointment regarding concurrent alcohol use (p.12-13).

---

## [Decision Letter · Decision Letter 1]

9 Sep 2022

PONE-D-21-30439R1Active versus sham transcranial direct current stimulation (tDCS) as an adjunct to varenicline treatment for smoking cessation: study protocol for a double-blind single dummy randomized controlled trialPLOS ONE

Dear Dr. Zawertailo,

Thank you for submitting your manuscript to PLOS ONE. After careful consideration, we feel that it has merit but does not fully meet PLOS ONE’s publication criteria as it currently stands. Therefore, we invite you to submit a revised version of the manuscript that addresses the points raised during the review process.

We look forward to receiving your revised manuscript.

Kind regards,

Kabirullah Lutfy

Guest Editor

PLOS ONE

Journal Requirements:

Additional Editor Comments:

Dear Dr. Zawertailo,

I served as a reviewer on the original version of your manuscript and requested you to add the varenicline group alone, but I was later assigned to serve as a guest editor on your revised versions. Please make the necessary changes, as I emailed you yesterday, and send us the further revised manuscript addressing the changes requested.

I look forward to receiving your further revised manuscript at your earliest convenience.

Sincerely,

Reviewers' comments:

Reviewer's Responses to Questions

**Comments to the Author**

1. Does the manuscript provide a valid rationale for the proposed study, with clearly identified and justified research questions?

Reviewer #1: Yes

Reviewer #3: Yes

Reviewer #4: Yes

2. Is the protocol technically sound and planned in a manner that will lead to a meaningful outcome and allow testing the stated hypotheses?

Reviewer #1: Yes

Reviewer #3: Yes

Reviewer #4: Yes

3. Is the methodology feasible and described in sufficient detail to allow the work to be replicable?

Reviewer #1: Yes

Reviewer #3: Yes

Reviewer #4: Yes

4. Have the authors described where all data underlying the findings will be made available when the study is complete?

Reviewer #1: Yes

Reviewer #3: Yes

Reviewer #4: Yes

5. Is the manuscript presented in an intelligible fashion and written in standard English?

Reviewer #1: Yes

Reviewer #3: Yes

Reviewer #4: Yes

6. Review Comments to the Author

You may also provide optional suggestions and comments to authors that they might find helpful in planning their study.

Reviewer #1: In this article Zawertailo and colleagues explored the effectiveness of tDCS in combination with varenicline for smoking cessation. I really appreciate the Authors’ response to the points I have raised in the first round of review, as well as their clarifications to some of my concerns.

I only have few last suggestions to do, to further improve the theoretical background of the present paper and its argumentation: in this regard, I would recommend deepening the information about the application of Non-invasive brain stimulation in modulating cortical excitability, and how these techniques can be used to investigate altered brain circuits (i.e., primary sensorimotor cortex, dorsolateral prefrontal cortex and the midbrain nucleus cuneiformis) in chronic smokers (https://doi.org/10.3390/biomedicines9070734;
https://doi.org/10.3390/biomedicines10030627;
https://doi.org/10.1016/j.tins.2022.04.003;
https://doi.org/10.1111/psyp.14122;
https://doi.org/10.1183/13993003.00362-2019).

Overall, this is a timely and needed study, and I look forward to seeing further studies on this issue by these authors in the future.

Thank You for your work.

Reviewer #3: The authors have done a very thorough job in detailing and addressing the review comments by the reviewers of the previous version of this manuscript.

The authors have adequately addressed all of my previous comments and concerns.

I have no further commentary or suggestions.

Reviewer #4: This is a proof of concept trial, therefore the statistical content is minimal, but seems to be planned well. The other reviewers have given very detailed comments--I was not a reviewer in the first round. The sample size, randomization, blinding, and statistical analysis plan are sound.

7. PLOS authors have the option to publish the peer review history of their article (what does this mean?). If published, this will include your full peer review and any attached files.

Reviewer #1: No

Reviewer #3: **Yes: **Joshua S. Rodefer

Reviewer #4: No

---

## [Author Response · Author response to Decision Letter 1]

11 Sep 2022

Reviewer: "I only have a few last suggestions to do, to further improve the theoretical background of the present paper and its argumentation: in this regard, I would recommend deepening the information about the application of Non-invasive brain stimulation in modulating cortical excitability, and how these techniques can be used to investigate altered brain circuits (i.e., primary sensorimotor cortex, dorsolateral prefrontal cortex and the midbrain nucleus cuneiformis) in chronic smokers

Immune Influencers in Action: Metabolites and Enzymes of the Tryptophan-Kynurenine Metabolic Pathway

(https://urldefense.com/v3/__https://doi.org/10.3390/biomedicines9070734__;!!FxkXuJIC!fV9JcA4KFOK2ywkccMQl4FPpsWA5s2JJsDlTtOnp-ZcXs30CyE0fE9YrN_A0u189lpM-Rd9dKTNrOF1887PKWiw$ ; 

The Neurobiological Correlates of Gaze Perception in Healthy Individuals and Neurologic Patients

https://urldefense.com/v3/__https://doi.org/10.3390/biomedicines10030627__;!!FxkXuJIC!fV9JcA4KFOK2ywkccMQl4FPpsWA5s2JJsDlTtOnp-ZcXs30CyE0fE9YrN_A0u189lpM-Rd9dKTNrOF18GgbK_1A$ ;

Functional interplay between central and autonomic nervous systems in human fear conditioning

https://urldefense.com/v3/__https://doi.org/10.1016/j.tins.2022.04.003__;!!FxkXuJIC!fV9JcA4KFOK2ywkccMQl4FPpsWA5s2JJsDlTtOnp-ZcXs30CyE0fE9YrN_A0u189lpM-Rd9dKTNrOF18966nyuM$ ; 

Characterizing cardiac autonomic dynamics of fear learning in humans

https://urldefense.com/v3/__https://doi.org/10.1111/psyp.14122__;!!FxkXuJIC!fV9JcA4KFOK2ywkccMQl4FPpsWA5s2JJsDlTtOnp-ZcXs30CyE0fE9YrN_A0u189lpM-Rd9dKTNrOF18e9b52MY$ ; 

Altered neural activity in brain cough suppression networks in cigarette smokers

https://urldefense.com/v3/__https://doi.org/10.1183/13993003.00362-2019__;!!FxkXuJIC!fV9JcA4KFOK2ywkccMQl4FPpsWA5s2JJsDlTtOnp-ZcXs30CyE0fE9YrN_A0u189lpM-Rd9dKTNrOF18FQfQne4$ )."

Response: Thank you for your comment. A paragraph has been added on p. 8 of the manuscript detailing more information on the potential of tDCS modulating cortical excitability and altered brain circuits.

---

## [Decision Letter · Decision Letter 2]

27 Oct 2022

Active versus sham transcranial direct current stimulation (tDCS) as an adjunct to varenicline treatment for smoking cessation: study protocol for a double-blind single dummy randomized controlled trial

PONE-D-21-30439R2

Dear Dr. Zawertailo,

We’re pleased to inform you that your manuscript has been judged scientifically suitable for publication and will be formally accepted for publication once it meets all outstanding technical requirements.

Kind regards,

Kabirullah Lutfy

Guest Editor

PLOS ONE

Additional Editor Comments (optional):

Reviewers' comments:

Reviewer's Responses to Questions

**Comments to the Author**

1. Does the manuscript provide a valid rationale for the proposed study, with clearly identified and justified research questions?

Reviewer #1: Yes

2. Is the protocol technically sound and planned in a manner that will lead to a meaningful outcome and allow testing the stated hypotheses?

Reviewer #1: Yes

3. Is the methodology feasible and described in sufficient detail to allow the work to be replicable?

Reviewer #1: Yes

4. Have the authors described where all data underlying the findings will be made available when the study is complete?

Reviewer #1: Yes

5. Is the manuscript presented in an intelligible fashion and written in standard English?

Reviewer #1: Yes

6. Review Comments to the Author

You may also provide optional suggestions and comments to authors that they might find helpful in planning their study.

Reviewer #1: The authors did an excellent job clarifying the questions I have raised in my previous round of review. Currently, this paper entitled ‘Active versus sham transcranial direct current stimulation (tDCS) as an adjunct to varenicline treatment for smoking cessation: study protocol for a double-blind single dummy randomized controlled trial’ is a well-written, timely piece of research and provides a useful summary of the existing status of knowledge of the effectiveness of tDCS in combination with varenicline for smoking cessation.

Overall, this is a timely and needed work. It is well researched and nicely written, with a good balance between descriptive and narrative text.

I believe that this paper does not need a further revision, therefore the manuscript meets the Journal’s high standards for publication.

Thank You for your work.

7. PLOS authors have the option to publish the peer review history of their article (what does this mean?). If published, this will include your full peer review and any attached files.

Reviewer #1: No

---

## [Editor Report · Acceptance letter]

6 Nov 2022

PONE-D-21-30439R2 

Active versus sham transcranial direct current stimulation (tDCS) as an adjunct to varenicline treatment for smoking cessation: study protocol for a double-blind single dummy randomized controlled trial 

Dear Dr. Zawertailo:

I'm pleased to inform you that your manuscript has been deemed suitable for publication in PLOS ONE. Congratulations! Your manuscript is now with our production department. 

Kind regards, 

on behalf of

Professor Kabirullah Lutfy 

Guest Editor

PLOS ONE